# Synthesis, Structural Characterization, Hirschfeld Surface Analysis and Photocatalytic CO_2_ Reduction Activity of a New Dinuclear Gd(III) Complex with 6-Phenylpyridine-2-Carboxylic Acid and 1,10-Phenanthroline Ligands

**DOI:** 10.3390/molecules28227595

**Published:** 2023-11-14

**Authors:** Li-Hua Wang, Xi-Shi Tai

**Affiliations:** 1College of Biology and Oceanography, Weifang University, Weifang 261061, China; 2College of Chemistry and Chemical Engineering, Weifang University, Weifang 261061, China

**Keywords:** 6-phenylpyridine-2-carboxylic acid, 1,10-phenanthroline, dinuclear Gd(III) complex, synthesis, crystal structure, hirschfeld surface analysis, photocatalytic CO_2_ reduction

## Abstract

A new dinuclear Gd(III) complex was synthesized and named [Gd_2_(L)_4_(Phen)_2_(H_2_O)_2_(DMF)_2_]·2H_2_O·2Cl (**1**). Here, L is the 6-phenylpyridine-2-carboxylate anion, Phen represents 1,10-phenanthroline, DMF is called *N,N*-dimethylformamide, and Cl^−^ is the chloride anion, which is characterized by IR and single crystal X-ray diffraction analysis. The structural analysis reveals that complex (**1**) is a cation–anion complex, and each Gd(III) ion is eight-coordinated with four O atoms (O1, O5, O2a, O4a, or O1a, O2, O4, O5a) of four different bidentate L ligands, two O atoms (O6, or O6a) of DMF molecules, two N atoms (N1, N2, or N1a, N2a) of Phen ligands, and two O atoms (O3 or O3a) of coordinated water molecules. Complex (**1**) forms the three-dimensional π–π stacking network structure with cavities occupied by chloride anions and uncoordinated water molecules. The Hirschfeld surface of the complex (**1**) shows that the H···H contacts represented the largest contribution (48.5%) to the Hirschfeld surface, followed by C···H/H···C and O···H/H···O contacts with contributions of 27.2% and 6.0%, respectively. To understand the electronic structure of the complex (**1**), the DFT calculations have been performed. The photocatalytic CO_2_ reduction activity shows complex (**1**) has excellent catalytic activity with yields of 22.1 μmol/g (CO) and 6.0 μmol/g (CH_4_) after three hours. And the selectivity of CO can achieve 78.5%.

## 1. Introduction

The increased human social activities not only accelerate the consumption of fossil energy but also result in a surge in CO_2_ concentration in the atmosphere. Therefore, the catalytic conversion and utilization of CO_2_ have become a research hotspot. At present, studies on photocatalytic CO_2_ reduction have received extensive attention [1]. So far, many studies have reported that precious metal catalysts show high catalytic activity and selectivity in the photocatalytic CO_2_ reduction reaction [2,3,4,5,6,7]. However, the disadvantages of expensive costs and a lack of storage limit the further applications of precious metal catalysts. It is a trend to explore new-type photocatalysts. Recently, some metal complex photocatalysts have become one of the research hotspots due to their excellent properties in photocatalytic CO_2_ reduction [8,9,10,11,12,13,14]. However, their activities are still low and cannot meet the needs of industrial applications. Rare earth elements often exhibit many special activities due to their special electronic structures. Therefore, earth-based rare complex photocatalysts are likely to exhibit good activity in photocatalytic CO_2_ reduction. And there are few reports on THE photocatalytic CO_2_ reduction in rare earth metal complexes [15,16].

Herein, we have chosen the Gd elements as the research object and prepared a new dinuclear Gd(III) complex using GdCl_3_·6H_2_O, 6-phenylpyridine-2-carboxylic acid, 1,10-phenanthroline, and NaOH as reactants. The complex (**1**) was characterized by IR and single crystal X-ray diffraction analysis, thereby confirming it is a cation–anion complex, and each Gd(III) ion is eight-coordinated with four O atoms. Subsequently, the photocatalytic CO_2_ reduction activity of complex (**1**) has been explored and found to have excellent catalytic activity, with yields of 22.1 μmol/g (CO) and 6.0 μmol/g (CH_4_) after three hours. And the selectivity of CO can achieve 78.5%. The synthetic route for complex (**1**) is shown in Appendix A.

## 2. Results and Discussion

### 2.1. Infrared Spectra

The infrared spectrum of complex (**1**) is given in Figure 1. The 6-phenylpyridine-2-carboxylic acid ligand exhibited characteristic bands at ca. 1646 (ν_as_COO^−^) and 1575 (ν_s_COO^−^) cm^−1^ [17], and in complex (**1**), they appeared at ca. 1619 (ν_as_COO^−^), and 1425 (ν_s_COO^−^) cm^−1^, respectively. Phen ligand showed characteristic bands at 1597 (C=N) cm^−1^ [18], and in complex (**1**), it appeared at 1577 (C=N) cm^−1^. These results indicate that the 6-phenylpyridine-2-carboxylic acid ligand and Phen ligand are coordinated with the Gd(III) ion. The IR results are consistent with single-crystal X-ray diffraction measurements of complex (**1**).

### 2.2. Structural Description of Complex *(**1**)*

The coordination environment of Gd(III) in complex (**1**) is given in Figure 2. Selected bond lengths (Å) and angles (°) for complex (**1**) are given in Table 1. Figure 3 shows the 3D network structure of complex (**1**). The asymmetric unit of complex (**1**) contains one Gd(III) ion, two 6-phenylpyridine-2-carboxylate ligands, one 1,10-phenanthroline ligand, one coordinated N,N-dimethylformamide molecule, one coordinated water molecule, one uncoordinated chloride anion, and one uncoordinated water molecule (Figure 2). The structural analysis reveals that complex (**1**) is a cation–anion complex and two Gd(III) ions are eight-coordinated with four O atoms (O1, O5, O2a, O4a, or O1a, O2, O4, O5a) of four different bidentate L ligands, one O atom (O6, or O6a) of DMF molecules, two N atoms (N1, N2, or N1a, N2a) of Phen ligands, and one O atom (O3 or O3a) of coordinated water molecules (symmetry code: 0.5-*x*, 1.5-*y*, 1.5-*z*). Complex (**1**) forms a dinuclear structure by bidentate chelate coordination mode of 6-phenylpyridine-2-carboxylate ligands, and the distance of two adjacent Gd(III) ions is 4.388 Å. Eight oxygen atoms are coordinated with two adjacent Gd(III) ions to form two stable eight-membered rings: ring 1 (O1-Gd1-O2a-C36a-O1a-Gd1a-O2-C36-O1) and ring 2 (O4a-Gd1-O5-C24-O4-Gd1a-O5a-C24a-O4a). The dihedral angle of ring one and ring two is 87.32°, indicating that the two eight-membered rings are nearly vertical. The bond distances of Gd–O and Gd–N are 2.313(3) Å (Gd1-O1), 2.379(3) Å (Gd1-O2a), 2.436(3) Å (Gd1-O3), 2.372(3) Å (Gd1-O4a), 2.331(3) Å (Gd1-O5), 2.406(3) Å (Gd1-O6), 2.560(3) Å (Gd1-N1), and 2.585(3) Å (Gd1-N2), respectively, which are consistent with those reported in the literature [14,19,20]. The uncoordinated chloride anion and the uncoordinated water molecule are embedded in the molecule by intramolecular O-H···Cl hydrogen bonds. And complex (**1**) molecules form a three-dimensional network structure (Figure 3) by the π–π interaction of aromatic rings.

Hydrogen bonds and π–π interaction play important roles in forming the 3D supermolecule of complex (**1**), and the detailed parameters are listed in the following Table 2 and Table 3.

### 2.3. DFT Computation

In order to understand the electronic structure of the complex, DFT calculations were performed. Figure 4 shows the electron density distributions and energy levels (eV) of HOMO-1, HOMO, LUMO, and LUMO + 1 for the ligands L and phen. These two ligands, neutral 6-phenylpyridine-2-carboxylic acid (L) and phen, were optimized at the theoretical level of B3LYP/6–31G* with the Gaussian 16 package [21,22,23]. In contrast to the planar phen, there is a dihedral angle of 15.5⁰ between the phenyl group and pyridine subunit in the ligand L, which is different from those of the ligands L in the complex (**1**) in the crystal (27.4⁰ and 8.9⁰). It indicates that the coordinates and steric hindrance in complex (**1**) change the planarity of the ligand L. Moreover, the electron density distributions and energy levels of the frontier molecular orbitals are shown in Figure 4, which were realized by the VMD package and the Multiwfn program [24].

### 2.4. Hirschfeld Surface Analysis of Complex (**1**)

The Hirschfeld surface of the complex (**1**) was analyzed by the CrystalExplorer software. As shown in Figure 5, the original crystal structure unit, the Hirschfeld surfaces, are mapped over the dnorm, di, and de of the crystal (Figure 5a–d). The two-dimensional (2D) fingerprint plots represented overall, and the top three interactions (H···H, C···H/H···C, and O···H/H···O) were shown in (Figure 5e–h). Based on the calculations, it can be concluded that the H···H contacts represented the largest contribution (48.5%) to the Hirschfeld surface, followed by C···H/H···C and O···H/H···O contacts with contributions of 27.2% and 6.0%, respectively. It is worth noting that the π–π stacking interactions play a subordinate role in forming the crystal for the C···C contacts with a Hirschfeld surface contribution percentage of 5.1%.

### 2.5. Photocatalytic CO_2_ Reduction Activity Assessment of the Complex *(**1**)*

As a newly discovered Gd(III) complex material, we want to try to explore its application field. Therefore, we tested its photocatalytic CO_2_ reduction activity, and the results are shown in Figure 6. Figure 6(a) shows that photocatalytic CO_2_ reduction activity using complex (**1**). Figure 6(b) describes that product selectivity diagram in the photocatalytic CO_2_ reduction reaction. It can be observed that the Gd(III) complex exhibits obvious photocatalytic CO_2_ activity. The main products of the whole photocatalytic reaction are CO and CH_4_, and their yields have reached 22.1 μmol/g and 6.0 μmol/g, respectively, after three hours of UV–vis light irradiation. In addition, the selectivity of CO can achieve 78.5%. With the increase in reaction times, the total amount of product also gradually increased, indicating that the photocatalytic CO_2_ reduction was sustainable. We have reported that a new Gd(III) coordination polymer exhibited photocatalytic CO_2_ reduction with a CO yield of 60.3 μmol•g^−1^ and a CO selectivity of 100% [16]. Compared with our previous results, complex (**1**) gives a different product, activity, and selectivity in photocatalytic CO_2_ reduction. These results demonstrate that the environment of the catalytic center Gd(III) is vital to its catalytic activity. As a photocatalyst, the light absorption capacity is important. So, the UV–vis absorption spectrum of complex (**1**) was examined. Appendix A exhibits the UV–vis absorption spectrum of the complex (**1**). It could be observed that the absorption edge of complex (**1**) is in the range of ultraviolet. Therefore, the researchers can design an idea to expand the light absorption capacity of complex (**1**) to improve its performance in photocatalytic CO_2_ reduction in future studies.

## 3. Experimental

### 3.1. Materials and Measurements

The materials of GdCl_3_·6H_2_O, 6-phenylpyridine-2-carboxylic acid, 1,10-phenanthroline, and NaOH were used as received from Jilin Chinese Academy of Sciences-Yanshen Technology Co., Ltd (Changchun, China). IR spectra were recorded on a Tianjin Gangdong FTIR-850 spectrophotometer (KBr discs, range 4000~400 cm^–1^). The Hirschfeld surface of the complex (**1**) was analyzed by the CrystalExplorer software 21.5 [25]. The crystal data of complex (**1**) were received on a Bruker CCD area detector (SuperNova, Dual, Cu at zero, 296.15 K, multi-scan).

### 3.2. Synthesis of Complex *(**1**)*

GdCl_3_·6H_2_O (0.1858 g, 0.5 mmol), 6-phenylpyridine-2-carboxylic acid (0.1992 g, 1.0 mmol), 1,10-phenanthroline (0.1802 g, 1.0 mmol), and NaOH (0.040 g, 1.0 mmol) were added to a 100 mL flask containing 30 mL of water–ethanol–DMF (*v:v:v* = 2:3:1) solution. The mixed suspension was stirred at 70 °C for 5 h and then cooled to room temperature. The colorless block crystals of complex (**1**) were obtained in four weeks.

### 3.3. Crystal Structure Determination

Single-crystal X-ray diffraction measurement of complex (**1**) was carried out on a Bruker CCD area detector and using Olex2 [26] for data collection at 219.98 (10) K. The structure was solved and refined with the SHELXT [17] and SHELXL [27] programs, respectively. The coordinates of hydrogen atoms were refined without any constraints or restraints. All non-hydrogen atoms were refined anisotropically. The hydrogen atoms were positioned geometrically (C–H = 0.93–0.96 Å and O–H = 0.85–1.06 Å). Their *U*_iso_ values were set to 1.2 *U*_iso_ or 1.5 *U*_iso_ of the parent atoms. Crystallographic data and structural refinement details of complex (**1**) are summarized in Table 4.

Crystallographic data for the structure reported in this paper have been deposited with the Cambridge Crystallographic Data Centre as supplementary publication No. CCDC 2292956. The CIF file can be obtained conveniently from the website: https://www.ccdc.cam.ac.uk/structures (accessed on 15 October 2023)

### 3.4. Photocatalytic CO_2_ Reduction Evaluation

The process of photocatalytic CO_2_ reduction is as follows: First, the 50 mg complex (**1**) sample was uniformly dispersed into 100 mL of deionized water H_2_O in a quartz reactor and sealed. The reaction temperature was kept at 20 °C using the cooling water circulation equipment. Subsequently, high-purity CO_2_ gas was bubbled into the above suspension solution with vigorous stirring for 15 min. Then, the reactor was irradiated by a 300 W Xe arc lamp (PLS-SXE300, Beijing Trusttech Co., Ltd., Beijing, China). The gas has been released every hour and tested via a gas chromatograph (FID detector, Shandong Huifen Instrument Co., Ltd., Laiwu, China).

## 4. Conclusions

In summary, a new dinuclear Gd(III) complex has been synthesized and characterized by IR and X-ray single-crystal diffraction analysis. The Hirschfeld surface of the complex (**1**) was analyzed. The photocatalytic CO_2_ reduction experiment showed that complex (**1**) has excellent catalytic activity with yields of 22.1 μmol/g (CO) and 6.0 μmol/g (CH_4_) after three hours. And the selectivity of CO can achieve 78.5%. It provides references for us to continue the study on the synthesis of rare earth metal complexes and their photocatalytic activities in the CO_2_ reduction reaction.

## Figures and Tables

**Figure 1 molecules-28-07595-f001:**
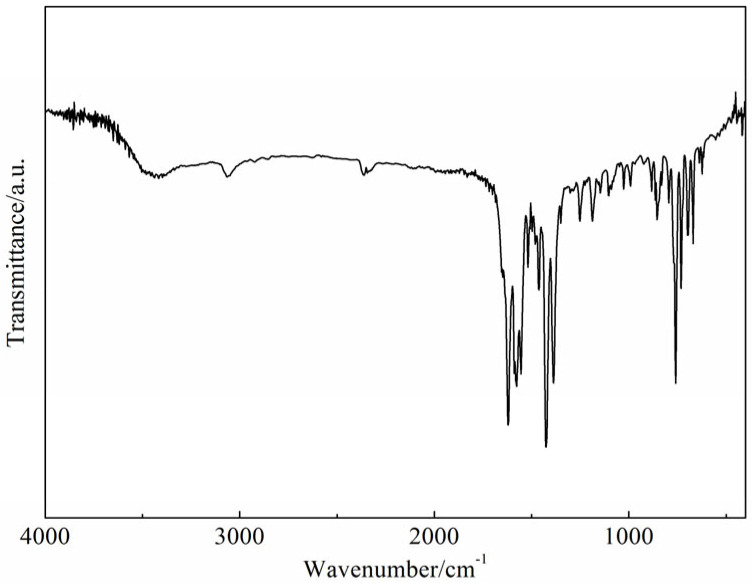
The infrared spectrum of the complex (**1**).

**Figure 2 molecules-28-07595-f002:**
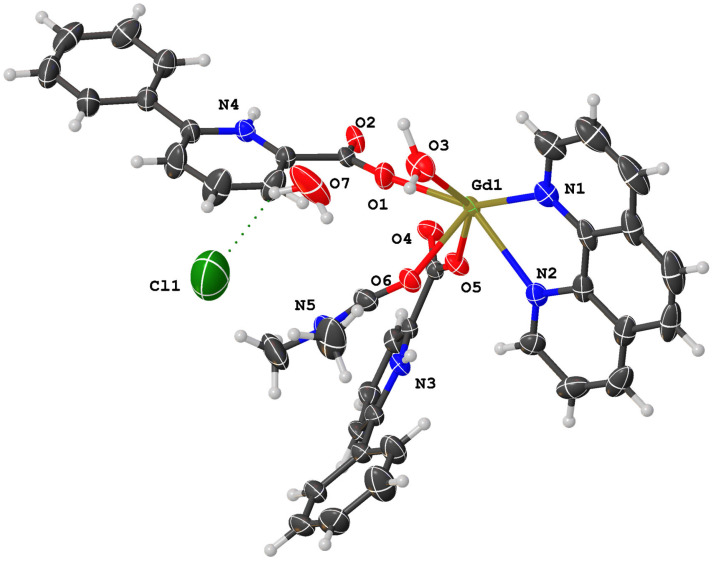
The coordination environment of Gd(III) in complex (**1**).

**Figure 3 molecules-28-07595-f003:**
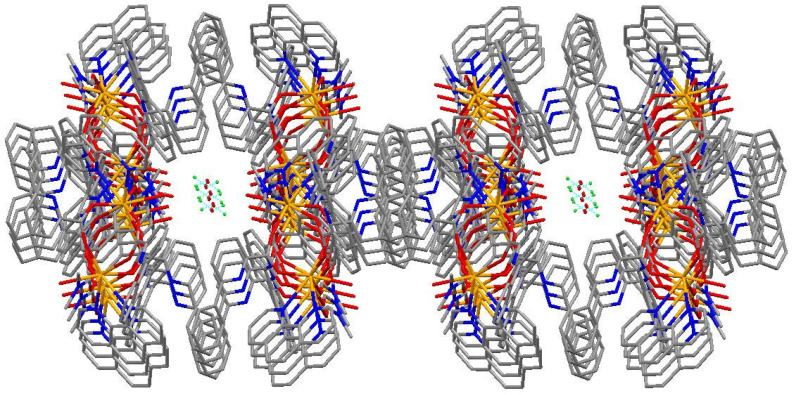
Three-dimensional network structure of the complex (**1**).

**Figure 4 molecules-28-07595-f004:**
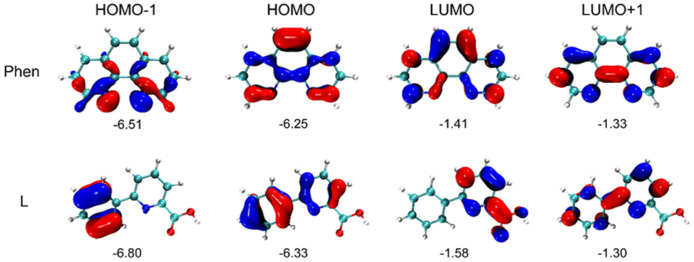
Electron density distributions and energy levels (eV) of HOMO-1, HOMO, LUMO, and LUMO+1 for the ligands L and phen (isovalue = 0.05 e•bohr-3). All energies are in eV.

**Figure 5 molecules-28-07595-f005:**
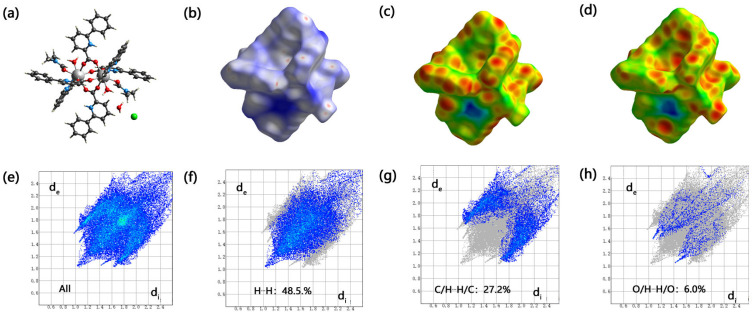
The Hirschfeld surface of the complex (**1**).

**Figure 6 molecules-28-07595-f006:**
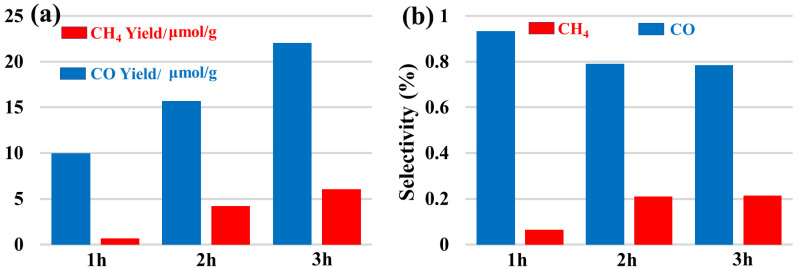
(**a**) Photocatalytic CO_2_ reduction activity using complex (**1**); (**b**) product selectivity diagram in the photocatalytic CO_2_ reduction reaction.

**Table 1 molecules-28-07595-t001:** Selected bond lengths (Å) and bond angles (°) for complex (**1**).

Bond	*d*	Angle	(°)
Gd1-N1	2.560(3)	N1-Gd1-N2	63.79(11)
Gd1-N2	2.585(3)	N1-Gd1-O1	147.23(12)
Gd1-O1	2.313(3)	O1-Gd1-N2	145.07(11)
Gd1-O2a	2.379(3)	O1-Gd1-O2a	123.07(11)
Gd1-O3	2.436(3)	O1-Gd1-O3	77.17(12)
Gd1-O4a	2.372(3)	O1-Gd1-O4a	79.20(10)
Gd1-O5	2.331(3)	O1-Gd1-O5	74.45(11)
Gd1-O6	2.406(3)	O1-Gd1-O6	81.61(11)
		N1-Gd1-O2a	74.76(11)
		O2a-Gd1-N2	69.35(10)
		O2a-Gd1-O3	139.53(10)
		O2a-Gd1-O6	139.90(10)
		N1-Gd1-O3	73.26(12)
		O3-Gd1-N2	115.87(12)
		N1-Gd1-O4a	79.48(11)
		O4a-Gd1-N2	134.52(10)
		O4a-Gd1-O2a	76.06(10)
		O4a-Gd1-O3	74.29(11)
		O4a-Gd1-O6	143.58(10)
		N1-Gd1-O5	138.32(11)
		N2-Gd1-O5	77.15(11)
		O5-Gd1-O2a	78.50(10)
		O5-Gd1-O3	141.59(11)
		O4a-Gd1-O5	123.93(10)
		O5-Gd1-O6	79.25(10)
		O6-Gd1-N1	101.73(11)
		N2-Gd1-O6	73.42(10)
		O6-Gd1-O3	71.41(10)

Symmetry transformations: a: 1/2 − *x*, −1 + *y*, 1 − *z*.

**Table 2 molecules-28-07595-t002:** Detailed parameters of hydrogen bonds in complex (**1**).

Donor-H	Acceptor	D-H (Å)	H^…^A (Å)	D^…^A (Å)	D-H^…^A (°)
O3-H3A	O7	1.07	2.46	3.125(6)	120
O3-H3B	O7 ^#1^	1.07	2.59	3.294(6)	123
O7-H7B	Cl1	0.85	2.50	3.143(5)	133

Symmetric operation code: ^#1^: 1/2 − *x*, *y*, 1 − *z*.

**Table 3 molecules-28-07595-t003:** Detailed parameters of π–π stacking interactions in complex (**1**).

Ring1	Ring2	Symmetry	Distance between Ring Centroids	Slippage
Cg2	Cg8	1/2 − *x*, 1/2 − *y*, 3/2 − *z*	3.713(2)	1.412
Cg5	Cg5		3.465(3)	0.620
Cg5	Cg8		3.746(3)	1.563
Cg5	Cg9		3.432(3)	0.583
Cg5	Cg10		3.464(2)	0.731
Cg8	Cg2		3.713(2)	1.577
Cg8	Cg5		3.746(3)	1.554
Cg8	Cg9		3.526(2)	1.002
Cg9	Cg5		3.432(3)	0.527
Cg9	Cg8		3.526(2)	0.963
Cg9	Cg9		3.800(2)	1.722
Cg9	Cg10		3.586(2)	1.172
Cg10	Cg5		3.465(2)	0.711
Cg10	Cg9		3.586(2)	1.186
Cg10	Cg10		3.6798(19)	1.445

Ring number: Cg2: N2-C6-C7-C10-C11-C12; Cg5: C4-C5-C6-C7-C8-C9; Cg8: N1-C1-C2-C3-C4-C9-C8-C7-C6-C5; Cg9: N2-C6-C5-C4-C9-C8-C7-C10-C11-C12; Cg10: N1-C1-C2-C3-C4-C9-C8-C7-C10-C11-C12-N2-C6-C5.

**Table 4 molecules-28-07595-t004:** Crystallographic data and structural refinement details of complex (**1**).

Empirical formula	C_78_H_74_Cl_2_Gd_2_N_10_O_14_
Formula weight	1760.87
Temperature/*K*	219.98(10)
Crystal system	monoclinic
Space group	*P*-1
*a*/Å	25.5838(6)
*b*/Å	13.6998(4)
*c*/Å	21.8202(6)
*α*/°	90
*β*/°	90.824(3)
*γ*/°	90
Volume/Å^3^	7647.0(4)
*Z*	4
ρ_calc_, mg/mm^3^	1.529
*μ*/mm^−1^	1.860
*S*	1.060
*F*(000)	3544
Index ranges	−23 ≤ h ≤ 30, −13 ≤ k ≤ 16, −25 ≤ l ≤ 25
Reflections collected	17406
2θ/°	4.722–49.999
Independent reflections	6746 [R(int) = 0.0252]
Data/restraints/parameters	6746/2/484
Goodness-of-fit on *F*^2^	1.044
Refinement method	Full-matrix least-squares on *F*^2^
Final *R* indexes [*I* ≥ 2σ (*I*)]	R_1_ = 0.0328, *wR*_2_ = 0.0792
Final *R* indexes [all data]	R_1_ = 0.0397, *wR*_2_ = 0.0747

## Data Availability

Data is contained within the article and Appendix A.

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
