# Peer review of "Synthesis, Structural Characterization, Hirschfeld Surface Analysis and Photocatalytic CO2 Reduction Activity of a New Dinuclear Gd(III) Complex with 6-Phenylpyridine-2-Carboxylic Acid and 1,10-Phenanthroline Ligands"

_molecules, 2023, doi:10.3390/molecules28227595_

Round 1

Reviewer 1 Report

Comments and Suggestions for Authors

The paper provides a good description of crystal synthesis, characterization, and performance evaluation. In terms of structure, it has also been well organized. Therefore, it can be accepted after revised following issues:

1. Hydrogen bonds and π-π interaction play important roles in forming 3D supermolecule of complex (1), however, the relevant data (like hydrogen bonds distance and angle, distance of π-π rings) are not listed in the manuscript. The author should provide these data in Tables.

2. The manuscript contained Chinese characters in section 3.2 (like the unit of 87.32), the author should check the manuscript carefully.

3. In the reference section, the DOI link of ref-23 is missing.

4. In the introduction section, the “BiOIBr” description is not appropriate, should be revised as “BiOIxBry”.

5. As a photocatalyst, the UV-VIS absorption spectrum of complex (1) should be examined to clarify its light absorption capacity.

Comments on the Quality of English Language

There are a few semantic ambiguities that should be revised. For example, the sentence 'The structure analysis reveals that complex (1) is a cation-anion complex and ...' should be modified to 'The structure analysis reveals that complex (1) is a cationic polymer and ...'.

Author Response

  1. Hydrogen bonds and π-π interaction play important roles in forming 3D supermolecule of complex (1), however, the relevant data (like hydrogen bonds distance and angle, distance of π-π rings) are not listed in the manuscript. The author should provide these data in Tables.

Response:

   “Hydrogen bonds and π-π interaction play important roles in forming 3D supermolecule of complex (1), the detail parameters are listed in the following Table 3 and Table 4” . These revisions have been highlighted in red in the text.

Table 3. Detailed parameters of hydrogen bonds in complex (1)

Donor-H

Acceptor

D-H (Å)

HA (Å)

DA (Å)

D-HA (°)

O3-H3A

O7

1.07

2.46

3.125(6)

120

O3-H3B

O7#1

1.07

2.59

3.294(6)

123

O7-H7B

Cl1

0.85

2.50

3.143(5)

133

Symmetric operation code: #1: 1/2-x, y, 1-z

Table 4. Detailed parameters of π-π stacking interactions in complex (1)

Ring1

Ring2

Symmetry

Distance between ring centroids

Slippage

Cg2

Cg8

1/2-x, 1/2-y, 3/2-z

3.713(2)

1.412

Cg5

Cg5

3.465(3)

0.620

Cg5

Cg8

3.746(3)

1.563

Cg5

Cg9

3.432(3)

0.583

Cg5

Cg10

3.464(2)

0.731

Cg8

Cg2

3.713(2)

1.577

Cg8

Cg5

3.746(3)

1.554

Cg8

Cg9

3.526(2)

1.002

Cg9

Cg5

3.432(3)

0.527

Cg9

Cg8

3.526(2)

0.963

Cg9

Cg9

3.800(2)

1.722

Cg9

Cg10

3.586(2)

1.172

Cg10

Cg5

3.465(2)

0.711

Cg10

Cg9

3.586(2)

1.186

Cg10

Cg10

3.6798(19)

1.445

Ring number:

Cg2: N2-C6-C7-C10-C11-C12; Cg5: C4-C5-C6-C7-C8-C9;

Cg8: N1-C1-C2-C3-C4-C9-C8-C7-C6-C5;

Cg9: N2-C6-C5-C4-C9-C8-C7-C10-C11-C12;

Cg10: N1-C1-C2-C3-C4-C9-C8-C7-C10-C11-C12-N2-C6-C5

  1. The manuscript contained Chinese characters in section 3.2 (like the unit of 87.32), the author should check the manuscript carefully.

Response:

We have revised the Chinese characters in the text.

  1. In the reference section, the DOI link of ref-23 is missing.

Response:

   We have added the DOI link of ref-23 as DOI: 10.14102/j.cnki.0254-5861.2013.10.022.

  1. In the introduction section, the “BiOIBr” description is not appropriate, should be revised as “BiOIxBry”

Response:

   We have revised the introduction. The “BiOIBr” description has been deleted.

  1. As a photocatalyst, the UV-VIS absorption spectrum of complex (1) should be examined to clarify its light absorption capacity.

Response:

Based on the reviewer's questions above, we have “As a photocatalyst, the light absorption capacity is important. So, the UV-VIS absorption spectrum of complex (1) was examined. Figure S1 exhibits the UV-vis absorption spectrum of the complex (1). It could be observed that the absorption edges of complex (1) is in the ultraviolet range. Therefore, the researchers can design an idea to expand the light absorption capacity of complex (1) for improving its performance of photocatalytic CO2 reduction in future studies.” in the text. And the .

Figure S1. The UV-vis absorption spectrum of the complex (1).

Reviewer 2 Report

Comments and Suggestions for Authors

This work needs more clarification, rewriting and addition of more experimental evidence to be considered for publication in the journal Molecules. Although, this paper would benefit more with a  complete rewrite but I am going to suggest some of the major issues:

1. An unimpressive, and incomplete introduction. Kindly, add more recent research work in the introduction section. Photocatalytic reduction of CO2 is an important research area,  and one can find so much relevant and latest information on searching. Kindly add new and recent literature.

There were instances of bad grammar and English throughout the manuscript. I am not going to point out all of such instances. I dont think the English of this paper merits to be published as is. I highly recommend seeking help of an English speaking colleague.

Also, remove 'ding et al'.... reported this.... or Taro et al synthesized.... to a better writing style.

2. Provide  detailed reaction schemes 1 containing the chemical structures of ligands, the synthesized complex, the reaction conditions (time, pH, temperature, solvent etc).

3. Abstract, line 3, Cl is chloride anion is a bad instance. It should be Cl-

4.  Section 2.4, What is  meant by '50 mg of the sample' what sample?

5. For section 2.4 as a supporting information provide all the gas chromatograms showing the products formed (CH4 and CO). And equally significant, provide blanks of the reactions meaning the solvent, substrate  (CO2) and gas chromatograms of these black solutions showing no organic contamination prior to experiments.

-Another blank experiment useful would be all the reagents irradiated with Xe arc lamp without the complex to show this results in the reduction of the CO2 and when this complex is absent, no reduction is seen.

6. For the same section, please also provide calibration curves along with methods of the yield calculations.

7.  In order to understand the electronic structures of the complex, please provide DFT (Density Functional Theory) Calculations.

8. Also provide 1H (proton) and 13C (carbon) NMR of the synthesized complex. It takes only minutes to a few hours to provide this important information but the results are comprehensive.

Comments on the Quality of English Language

This paper needs to be re-written. There are grammatical errors throughout the manuscript. Kindly improve the sentences, active and passive voice etc.

Author Response

  1. An unimpressive, and incomplete introduction. Kindly, add more recent research work in the introduction section. Photocatalytic reduction of CO2is an important research area, and one can find so much relevant and latest information on searching. Kindly add new and recent literature.There were instances of bad grammar and English throughout the manuscript. I am not going to point out all of such instances. I dont think the English of this paper merits to be published as is. I highly recommend seeking help of an English speaking colleague. Also, remove 'ding et al'.... reported this.... or Taro et al synthesized.... to a better writing style.

Response:

   We are very thankful for reviewer’s high value comments. We have carefully checked the main text and revised the introduction and some grammatical mistake in the main text. These revisions have been highlighted in red.

  1. Provide detailed reaction schemes 1 containing the chemical structures of ligands, the synthesized complex, the reaction conditions (time, pH, temperature, solvent etc).

Response:

We have drawn reaction schemes, and it has been shown in scheme S1.

Scheme S1. Synthetic route for complex (1)

  1. Abstract, line 3, Cl is chloride anion is a bad instance. It should be Cl-

Response:

   We have revised the Cl as Cl-.

  1. Section 2.4, What is meant by '50 mg of the sample' what sample?

Response:

   We have revised the sample as “complex (1)”, which was highlighted in red on page 3.

  1. For section 2.4 as a supporting information provide all the gas chromatograms showing the products formed (CH4and CO). And equally significant, provide blanks of the reactions meaning the solvent, substrate (CO2) and gas chromatograms of these black solutions showing no organic contamination prior to experiments.-Another blank experiment useful would be all the reagents irradiated with Xe arc lamp without the complex to show this results in the reduction of the CO2 and when this complex is absent, no reduction is seen.

Response:

Based on the reviewer's questions above, we have supplemented some photocatalytic CO2 reduction experiments. The gas chromatograms of black solution (before the reactor is irradiated by the Xe arc lamp) and third hour (the reactor was irradiated for three hours) in the photocatalytic CO2 reduction have been obtained via the screen capture, which has been shown in Figure R1. It could be observed that the gas chromatograms of black solution (Figure R1a) only exhibited a vibration signal peak at about 1.7 min caused by injecting sample gas. The result indicates before the entire photocatalytic CO2 reduction test is performed, there is no other pollution gas. After the photocatalytic CO2 reduction test was carried out for three hours, there are two new peaks in the gas chromatograms (Figure R1b) to be observed, which is CO (at 2.5 min) and CH4 (at 6.2 min), respectively. In addition, we have also performed the photocatalytic CO2 reduction test without the complex, no any product was detected (Figure R2).

Figure R1. The gas chromatograms of black solution (a) and third hour (b) in the photocatalytic CO2 reduction.

Figure R2. The gas chromatograms of photocatalytic CO2 reduction without the complex.

  1. For the same section, please also provide calibration curves along with methods of the yield calculations.

Response:

The method of the yield calculation is following:

1mL of the gas has been got out every hour and tested via a gas chromatograph. The peak area of the product was obtained. Subsequently, we quantified the number of moles of per peak area via using the CO and CH4 standard gases, which is noted as Y1CO and Y1CH4, respectively. The total amount of CO and CH4 is calculated according to the equation below.

  and

Sco and SCH4 is the peak areas of CO and CH4, which was measured using the gas chromatograph. Sr is the reactor volume. The related above content has been added into the supporting information.

  1. In order to understand the electronic structures of the complex, please provide DFT (Density Functional Theory) Calculations.

Response:

we have supplemented DFT Computation as following:

   3.3.  DFT Computation

   To understand the electronic structure of the complex, the DFT calculations were performed. The two ligands, netural 6-phenylpyridine-2-carboxylic acid (L) and phen were optimized at the theoretical level of B3LYP/6-31G* with Gaussian 16 package [24-26]. In contrast to the planar phen, there has a dihedral angle of 15.5⁰ between phenyl group and pyridine subunit in the ligand L, which is different from those of the ligands L in the complex (1) in the crystal (27.4⁰ and 8.9⁰). It indicates that the coordinate and steric hidrance in the complex (1) change the planarity of the ligand L. Moreover, the electron density distributions and energy levels of the frontier molecular obitals were shown in Figure 4, which were realized by VMD package and Multiwfn program [27].

Figure 4. Electron density distributions and energy levels (eV) of HOMO-1, HOMO, LUMO and LUMO+1 for the ligands L and phen (isovalue = 0.05 e•bohr-3) . All enegies are in eV.

  1. Also provide 1H (proton) and 13C (carbon) NMR of the synthesized complex. It takes only minutes to a few hours to provide this important information but the results are comprehensive.

Response:

The structure of complex (1) has been determined by Single-crystal X-ray diffraction , whcih determines the complex structure with greater accuracy than other spectroscopic methods. However, it also provides very useful suggestions for our future studies.

Round 2

Reviewer 2 Report

Comments and Suggestions for Authors

 1. I appreciate authors efforts to incorporate my ideas. However, I am not sure if my suggestion to add the gas chromatograms (blank containing only the reactants, chromatograms of the reaction without catalyst and the reaction catalyzed by the synthesized complex) was taken into consideration . The authors mention in their author's response letter that they have included the requested chromatograms as R1 and R2 but I cannot find the chromatograms. These are neither is the main text, nor in the supplementary section. I regret to say that I did not find such information. I cannot assess the revision of this manuscript completely without the requested chromatograms. These gas chromatograms are briefly mentioned (lines 88-89) but are given nowhere. Without concrete evidence, it is hard to conclude whether CO2 reduction occurred or not.

Please provide proper (professional grade) GC spectra for example see paper: Gas Chromatographic Separation of Carbon Dioxide, Carbon Oxysulfide, Hydrogen Sulfide, Carbon Disulfide, and Sulfur Dioxide.

2. Furthermore I see that 'method' of yield calculation but without THE CALIBRATION CURVES.  This should have been provided with the SI.

3. What is this newly added figure S-1 its axis should be revised to show the starting point of the curve. Figure should also have λmax.

4. There are still so many grammatically incorrect instances. I am going to mention only a few:

1. The abstract reads as "DFT calculations were carried out". This sentence appeared to be incoherent and seemed to have come out of nowhere.

2. There are so many inconsistencies related to the 'past' or 'present' tenses.

3. What is the meaning of "each Gd(III) ion is EIGHT COORDINATED"? Please revise this.

4. There are errors even in the re-written very first sentence of the introduction (red lines).

I suggest authors to take their time. Probably, if possible request a proper (sufficient) time to revise this manuscript to make it upto the mark of the journal 'Molecules'.

Comments on the Quality of English Language

Poorly written paper!

Author Response

Dear Editors and Reviewers:

Thanks for your comments concerning our manuscript entitled “Synthesis, Structural Characterization and Photocatalytic CO2 Reduction Activity of a New Dinuclear Gd(III) Complex with 6-Phenylpyridine-2-carboxylic acid and 1,10-Phenanthroline Ligands” (molecules-2700649). Those comments are all valuable and helpful for revising and improving our paper. We have carefully studied the comments and made corrections based on your comments. Modified portions are highlighted in red in the revised version. As suggested, we improved the manuscripts by improving the writing and adding more data and analyses. The revised manuscript is responded to, point by point.

Thank you and best regards.

Yours sincerely,

Corresponding author:       taixs@wfu.edu.cn.